# FOUNDATION MODELS FOR AGRICULTURAL SCIENCES - CHALLENGES AND OPPORTUNITIES

**Ioannis N. Athanasiadis**[*]
Artificial Intelligence Group
Wageningen University and Research
Wageningen, 6700AA, The Netherlands
`ioannis.athanasiadis@wur.nl; http://www.wur.ai`

## ABSTRACT

Foundation models, developed with self-supervised methods using unlabeled and multimodal multidisciplinary corpora, offer a new pathway for AI-driven discoveries in agricultural sciences. In this paper, we identify key challenges inherent to AI research in agricultural sciences, and advocate that an emerging new generation of agricultural AI, in the form of foundation models is necessary to kickstart data-driven discoveries in agricultural sciences. We argue that foundation models have the potential to act as jumpboards and enable a new generation of AI models for agricultural sciences, offering power and adaptability to accelerate innovation. By reflecting on agricultural science challenges, data, and theory, we identify pathways for model development and evaluation that will enable agricultural researchers to build on shared foundations instead of training new models from scratch for each application and location.

## 1 INTRODUCTION

Recent advances in artificial intelligence (AI) are changing fast the way we conduct research in agricultural sciences, and address global challenges related to food security. For example, computer vision, together with in-situ sensing technologies, are successfully addressing long-standing challenges in plant phenotyping: High-throughput phenotyping has now become mainstream, enabling a new generation of scientific discoveries founded on continuous, AI-driven measurements of plant growth (Tsaftaris & Scharr, 2019; Ubbens et al., 2025). Similarly, AI advances in Earth Observation (EO) and Remote Sensing (RS) have now been translated into impactful world-wide products for crop monitoring (van Tricht et al., 2023), offering a new level of food security insights that reaches a broad variety of stakeholders, from farmers to policymakers. Likewise, agricultural extension (i.e., farmer advice and education) is transforming because of the opportunities arising from Large Language Models (LLMs), enabling local language communication and widespread access to agronomy knowledge and education (Tzachor et al., 2023; Kpodo et al., 2024).

Despite a first wave of success, there has been relatively less impact to agricultural and food sciences from AI advances, especially when compared to other scientific domains and global challenges, such as biodiversity conservation and climate change (Duede et al., 2024). In part, this is because agriculture is inherently a complex, multifaceted problem: it concerns not only plants and animals growing in, and interacting with, their natural environment; it also concerns humans who intervene in these processes, as farmers, policymakers, technology providers, or consumers. Thus, agricultural research is commonly framed as the interaction of biological (genetic) material (G) with the natural environment (E), and humans who manage them (M), giving rise to complex, multi-disciplinary challenges across the G×E×M continuum.

In this work, we summarize key issues inherent to AI research in agricultural sciences, and advocate that an emerging new generation of agricultural AI, in the form of foundation models is necessary to kickstart data-driven discoveries in agricultural sciences. We argue that foundation models have the potential to act as jumpboards and enable a new generation of AI models for agricultural sciences, offering power and adaptability to accelerate innovation. By reflecting on agricultural science

---

[*]Thanks to the AgriScienceFM project consortium

challenges, data, and theory, we identify pathways for model development and evaluation that will enable agricultural researchers to build on shared foundations instead of training new models from scratch for each application and location.

## 2 AGRICULTURAL SCIENCES CHALLENGES

Agricultural systems study the interactions across humans, plants, animals and nature, which are both hard to capture and hard to understand. *Hard to capture*, as most experimental and observational datasets are (relatively) small, noisy, expensive and slow to collect, and offer only a partial view of the full interaction space. As an example, consider crop monitoring from remote sensing: In spite of their heavy influence, the effects of pests and diseases are rarely visible in satellite data until crop failure has occurred. *Hard to understand*, as agricultural sciences knowledge is fragmented across several disciplines, from genetics to agroecology and food systems science. As an example, most agronomy trials focus on farm management practices, and there is very small genetic or soil variation in these exposures. Together, these challenges have constrained the transformative potential of AI for agriculture: **Challenge 1**: complex human-plant-animal-environment interactions (G×E×M); **Challenge 2**: limited and fragmented data in misaligned corpora; and **Challenge 3**: siloed knowledge across several scientific disciplines.

Conventional machine and deep learning methods require large, annotated datasets to infer input-output relations from. Hence, their development for agriculture has been limited by the compartmentalization of agricultural research and data (across disciplines, modalities, and scales, see Fig.1), which means that available datasets are often too small, sparse and context-specific to effectively support the large-scale, data-hungry AI methods commonly used in other application domains. Therefore most AI deployments in agriculture fail to generalize in space and time, much more so when transferring to new species or production systems. The key challenge for agricultural AI is to develop models that can generalize effectively across the G×E×M continuum. AI models in agriculture need to demonstrate their skill in new environmental conditions, new genetic material, and new management regimes.

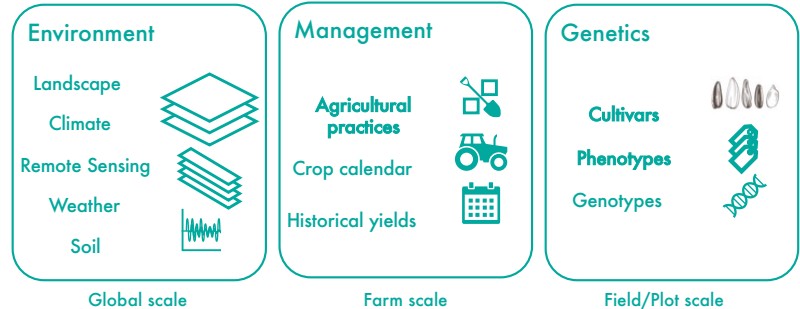

Figure 1: The agricultural data landscape is spans across sectors, modalities and scales

## 3 OPPORTUNITIES FOR AI

Agricultural sciences have accumulated diverse data across decades (e.g., field notebooks, UAV imagery, policy documents, extension bulletins), but these data are often unstructured, siloed or weakly annotated. For example, some of the largest open annotated datasets of crop breeding trials do not exceed 140K labelled plot cycles (Lopez-Cruz et al., 2023). This amount of data is orders of magnitude less than what conventional AI methods typically require to generalize effectively. Therefore, a key requirement for AI is to **build effective models with small amounts of multimodal corpora, which are typically both incomplete and not aligned** (Opportunity 1).

At the same time, agriculture is a very dynamic field. Historical data, even when available, are not representative enough for enabling effective generalization in future conditions. This not only driven by environmental factors, as climate change and the associated domain shifts, but also from a broad set of social and economic drivers that make humans improve agricultural practices. Even when

agricultural production systems remain the same for long time, the underlying agricultural systems are in constant change, including introducing new crop varieties, technology for crop protection, new milking robots or offering products to new markets. While such causal relations are relatively well understood as principles, key drivers of change are often not represented in the data available. Simulation models are not able to fill such gaps either. A second requirement for AI in agriculture is **how to regularize AI models with implicit knowledge** (Opportunity 2). How foundation models learn plausible representations from first principles rather from observations or simulation models alone, remains a relevant opportunity for agriculture-inspired AI research.

Last but not least, AI solutions in agriculture often suffer from overfitting to local effects, and typically require new data acquisition and additional labeling to transfer to new conditions and tasks. Activities that are are both slow and expensive. For example, to predict crop yield, annual crops result in a data acquisition speed of one label per year for each location. While this setting is not adequate for conventional, label-hungry, supervised AI approaches, foundation models building on top of large-scale, unlabeled and multi-modal corpora can provide a better starting point in downstream tasks with few labels. A third requirement for AI in agriculture is to **learn effective representations, in the form of causal priors, for a variety of downstream tasks, often formalized as few-shot learning problems** (Opportunity 3).

## 4 A WAY FORWARD FOR FOUNDATION MODELS IN AGRICULTURE

Foundation models, developed with self-supervised methods using unlabeled and multimodal multi-disciplinary corpora, offer a new pathway for AI-driven discoveries in agricultural sciences. Several general-purpose foundation models have become available in the past few years, demonstrating remarkable skill in a variety of computer vision or language-related downstream tasks, but also for scientific applications.

General-purpose foundation models have been applied to various agriculture-related tasks, mostly in computer vision ones (Li et al., 2024). A recent review on deep learning for plant phenotyping (Ubbens et al., 2025) underlines that general-purpose foundation models are underperforming despite the (initial) expectations, and that "*domain shift seems to remain one of the most pressing problems*". Similar are the findings from the use of general-purpose earth observation and weather foundation models in agriculture-related tasks, as phenology and crop yield estimation (Nedungadi et al., 2025), and LLMs for agricultural advisory and extension (Tzachor et al., 2022; Kpodo et al., 2024).

Recent work on adapting vision foundation models for assessing herbicide field trials in agriculture, has demonstrated performance improvements under domain shifts and low-annotation regimes in open field conditions (Benito-Del-Valle et al., 2025). Similarly holds for adapting a pretrained model for large scale crop type classification from earth observation data, that improves performance over conventional supervised methods (Butsko et al., 2025).

Agriculture-specific foundation models have the potential to accelerate scientific progress in agricultural research. By pre-training them on massive and heterogeneous unlabeled agricultural datasets, ranging from remote sensing images to weather, and to crop management and farm advisory data, and then adapted to highly specific tasks with limited additional data. This blend of power and adaptability will accelerate innovation, letting researchers build on shared foundations instead of training new models from scratch for each application and location. To this end, method development is necessary to focus on the opportunities identified above: (a) learning across small, multi-modal corpora with few paired examples (b) learn effective representations with implicit knowledge in the form of basic principles (c) learn effectively embeddings that represent the causal drivers of the underlying phenomena.

Foundation models can help overcome the challenges inherent to AI research in agriculture, in two distinct ways. First, by performance improvements in terms of accuracy or reduced uncertainty in decision-making, and, second, by lowering the data collection and annotation barriers for developing performant models in new conditions or tasks. Benchmarking the performance of foundation models in both ways is necessary for monitoring progress.

Currently, there is no standardized benchmarking framework for evaluating AI models, let alone foundation models, in agricultural sciences. The assessment of AI applications in agricultural re-

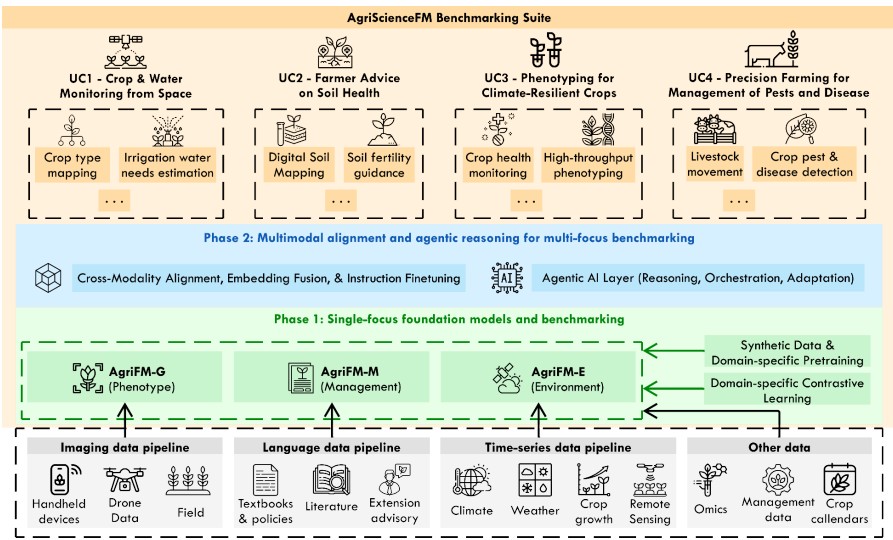

Figure 2: The AgriScienceFM project concept: Three distinct domain-specific foundation models are first trained and evaluated separately in single-focus benchmarks (Phase 1) to learn different sources of variation in a self-supervised way. Then the three models will further be aligned and orchestrated and evaluated in multi-focus evaluations (Phase 2).

search remains fragmented, ad hoc, and case-study-specific, making it difficult to compare models, interpret results, or measure progress (Li et al., 2024). Most AI studies in agriculture rely on a heterogeneous selection of datasets, with inconsistent separation of training and testing data across publications. Moreover, annotation quality varies significantly, reducing reproducibility and introducing potential biases that hinder fair comparison of different approaches. While some baseline studies have attempted to define benchmark tasks (Paudel et al., 2021; Zhou & Ryo, 2025; Paudel et al., 2025), their adoption as standard references by the broader community has been limited. While in computer vision-related tasks there are more established benchmarks (e.g. for crop classification, leaf counting), a unified evaluation framework with established data splits, metrics, and leaderboards is missing.

A benchmarking suite for AI in agriculture is necessary, not simply to measure progress and success, but to set ambitious goals addressing long-standing challenges in agricultural sciences and drive impact and AI innovation. Starting by identifying key gaps for AI in agricultural research, the selection and organization of downstream tasks is critical to frame adoption, adaption and expansion for future innovation. Design and development of corresponding benchmarks, including data, metrics and reference models needs to go beyond comparing only against naive baselines, or simple data-driven models. **Dual baselines** are necessary to measure progress that compare against both state of the art AI models and domain-specific ones.

Successful foundation models in other scientific fields like protein folding, climate forecasting, and material science rely on domain-specific models built on extensive scientific knowledge and standardized data. Protein folding, a long-standardized problem, now benefits from public data, metrics, and reference models, thanks to CASP competitions. We envision foundation models in agriculture designed and developed by incorporating agricultural knowledge and thoroughly benchmarked to prove their skill in challenging, domain-specific tasks against state-of-the-art agricultural methods. This is essential for measuring progress, reproducibility, and building trust in AI agricultural solutions.

Effective foundation model training should not only be evaluated with downstream benchmarks, but also by comparing with general-purpose foundation models. Agriculture-specific foundation models need to demonstrate their ability to generate high-quality agriculture-specific embeddings that capture complex relationships between agriculture drivers.

Unlike other scientific fields, agriculture is inherently multimodal and multiscale, lacking a single comprehensive theory and data layer for driving foundation model development. Instead, information needs to be compiled from several scientific domains and different scales across the G×E×M continuum, requiring know-how of several disciplines (e.g., agronomy, plant sciences, climate, geography, soil science, plant and animal genetics).

A way forward is to develop three complementary agriculture foundation models, each focusing on a distinct dimension of G×E×M, with the intention to effectively capture the causal drivers in each dimension, an approach we will follow in the European Project **AgriScienceFM**, that aspires to further AI foundation models for agricultural sciences. By concentrating each foundation model on a single dimension (G,E,M), we limit the scale of domain knowledge and relevant data types that are required for self-supervised pretraining. Each model encompasses the knowledge of one major driver of G×E×M interactions, and can be trained in a self-supervised manner with large scale unpaired data available. On top of each one of them it is feasible to build task-specific models, adaptable to several different applications, varying from crop monitoring from space, to farmer advisory, phenotyping resilient crops and precision farming.

Looking ahead, how the three models can be used together and evaluated in complex benchmarks that cut across the three G×E×M dimensions remains an open challenge for joint research. However we already envision a two tier-system, where in the first one there is a single major driver (G,E or M) and a single corresponding model is evaluated. In the second tier, multi-focus evaluations will include becnhmarking tasks where multiple drivers are interacting (GxExM). The three models can be either fused as modular experts working together for certain tasks, or be loosely orchestrated for more complex scientific reasoning. To this end, both cross-modality alignment and early enablers for agentic AI will be explored for complex agricultural research workflows. Recent advancements in uncertainty-aware embeddings, semantic enrichment, dynamic memory, and lightweight interoperability will be explored. This will offer insights, best practices, and reusable assets for future agricultural AI research.

In this paper, we motivated and outlined a conceptual framework for advancing in AI research for agricultural sciences, outlining how agriculture-specific foundation models support the long-term evolution of agricultural AI beyond stand-alone models, and small-scale testing and experimentation.

### ACKNOWLEDGMENTS

All twelve AgriScienceFM project partners have contributed to the shaping of this concept, during several online calls in the summer of 2025, while preparing the AgriScienceFM proposal for submission. I am deeply grateful for these discussions and looking forward for working together in the years to come. Thanks to the anonymous reviewers for their constructive feedback, that helped improving this manuscript and their recommendations for follow up.

This research has been selected for funding by the European Commission (AgriScienceFM). More about the project at http://www.agriscience.fm

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
