# OpenReview forum: "Foundation models for agricultural sciences - Challenges and opportunities"
_ICLR.cc/2026/Workshop/FM4Science — ICLR 2026 Workshop FM4Science Poster_

### Official Review · Reviewer_6nvk · 2026-02-15
**This work identifies key gaps in agricultural AI research and advocates for the development of foundation models tailored to the unique complexities of food systems**

**Rating:** 7
**Confidence:** 4

**Review:**

Overall the area is important and technical is reasonable and writing is clear. The work identifies that current datasets for crop trials are orders of magnitude smaller than the corpora used to train state-of-the-art LLMs, necessitating models that can learn from small, multimodal corpora with few paired examples

Strengths:
* Framework: Anchors AI development in the well-established GxExM scientific theory rather than generic AI leaderboards
* Realistic Assessment: Acknowledges the failure of general foundation models (like LLMs or general ViTs) in specialized agricultural tasks due to severe domain shift.
* Causal Prior Focus: Prioritizes embeddings that represent the underlying drivers of phenomena, which is essential for trustworthy decision-support systems.
* (Modular Scalability: Proposes a feasible architecture that limits the scale of required domain knowledge for any single model while allowing for complex reasoning through orchestrating modular experts.
* Benchmarking: Identifies the absence of a unified evaluation framework (similar to CASP for protein folding) as a primary barrier to progress.

Weakness:
* Purely Conceptual: The paper lacks empirical validation or a small-scale proof-of-concept of the proposed modular integration.
* Data Acquisition Barriers: While it advocates for models trained on massive heterogeneous datasets, it does not detail how to overcome the unstructured and siloed nature of existing field notebooks and UAV imagery.
* Orchestration Complexity: The paper leaves the mechanism of how these three models would be "fused" or "loosely orchestrated" as an open research challenge, providing few details on implementation

---

### Official Review · Reviewer_o9uN · 2026-02-20
**Agricultural sciences**

**Rating:** 6
**Confidence:** 3

**Review:**

This paper argues for building domain-specific foundation models for agricultural sciences. It uses the G×E×M framework to organize the problem and proposes three separate models for each dimension. It also calls for better benchmarking in the field. I found it a reasonable and interesting read overall.

Strengths

S1. The G×E×M framing is helpful. Organizing the problem around Genetics, Environment, and Management makes sense to me. It gives a clear reason for building three separate models rather than one big one. I found this more convincing than papers that just split by data modality.

S2. The benchmarking point feels important.The authors are right that agricultural AI lacks shared evaluation standards. The comparison to CASP in protein folding is a good analogy. I think this part could have real impact if the community picks it up.

S3. The paper is honest about its limitations.It does not claim to have solved anything. The three requirements it identifies feel grounded in real problems rather than made up to fit the proposal. I appreciated that.

Weaknesses

W1. The core proposal needs more detail.The three-FM idea is interesting but I was left unsure how it would actually work. There are no pretraining objectives, no fusion strategy, no architecture details. I would have liked at least a sketch of the approach.

W2. Some relevant recent work seems to be missing. I am not a deep expert in this area but I believe models like Prithvi-EO-2.0 and AgriFM are related to what the authors propose. Not discussing them makes it harder to understand what is new here.

W3. The benchmarking section stays too abstract. The authors call for standardized benchmarks but do not propose any specific tasks or metrics. I think even a rough list of candidate tasks would have made this more useful.

W4. It is unclear where the pretraining data comes from. The paper highlights that labeled agricultural data is very limited, with the largest datasets around 140K plot cycles. But then it assumes large unlabeled multimodal data exists for pretraining. I was not sure this tension was fully resolved. No specific datasets or data sources are named.

W5. There are no experiments at all. I understand this is a position paper but I think even a small preliminary result would help. Right now the claims about FM advantages over general-purpose models are not backed by any numbers.

---

### Official Review · Reviewer_DSVD · 2026-02-20
**Conceptual roadmap for agricultural foundation models lacking empirical validation and sufficient technical depth.**

**Rating:** 3
**Confidence:** 3

**Review:**

Quality: While the conceptual framework is sound, there are no experimental results, baseline comparisons, or proofs of concept to validate the proposed tripartite model strategy.
Clarity: The writing is clear, and Figure 1 effectively illustrates the multiscale nature of agricultural data. However, the technical details on how these three separate models would be aligned for complex reasoning remain vague and purely speculative.
Originality: The application of the paradigm to foundation model design is a thoughtful adaptation of classical agricultural science. However, the core idea, that domain-specific pre-training outperforms general-purpose models, is well-established in other scientific ML fields (e.g., climate or protein folding), making the conceptual contribution here incremental.
Significance: If fully realized, the proposed framework could have high significance for food security and high-throughput phenotyping. In its current form, however, it does not serve a functional contribution to the field.

Pros:
Relevant: Addresses the domain shift problem where general-purpose models (like standard LLMs or Vision Transformers) underperform in specialized agricultural tasks.
Structured Framework: The division of foundation models into Genetics (G), Environment (E), and Management (M) is a biologically grounded way to reduce the complexity of self-supervised learning.

Cons:
The main text is less than 4 full pages.
Lack of Empirical Results: There is a total absence of quantitative evaluation. A paper at this level should, at a minimum, provide a pilot study demonstrating how the joint embedding space outperforms a single unified model.
Theoretical Vaguery: The paper mentions uncertainty-aware embeddings and agentic AI as future directions, but provides no technical roadmap for how these complex components interact with the base foundation models.
No Benchmark Contribution: While the authors emphasize the need for a unified benchmarking suite, they do not propose one, which limits the paper’s immediate utility to other researchers.

---

### Meta-Review · Area_Chair_MqVq · 2026-03-02

**Recommendation:** Accept (Poster)
**Confidence:** 4

**Metareview:**

This submission has received three reviews. Two positive reviews with a "accept" and two "marginally above acceptance threshold". One reviewer rated the paper with a "reject".

After reading the reviews, I recommend this paper for "acceptance" and ask the authors to consider implementing the feedback given by all reviewers (especially reviewer DSVD) into the camera-ready version of the paper.

---

### Decision · Program_Chairs · 2026-03-03

Accept (Poster)